# Investigation of Neutrophil Extracellular Traps as Potential Mediators in the Pathogenesis of Non-Acute Subdural Hematomas: A Pilot Study

**DOI:** 10.3390/diagnostics12122934

**Published:** 2022-11-24

**Authors:** Michael T. Bounajem, Frederik Denorme, John L. Rustad, Robert A. Campbell, Ramesh Grandhi

**Affiliations:** 1Department of Neurosurgery, Clinical Neurosciences Center, University of Utah, Salt Lake City, UT 84112, USA; 2Program in Molecular Medicine, University of Utah, Salt Lake City, UT 84112, USA; 3Department of Internal Medicine, Division of General Medicine, University of Utah, Salt Lake City, UT 84112, USA; 4Department of Pathology, Division of Microbiology and Immunology, University of Utah, Salt Lake City, UT 84112, USA

**Keywords:** neutrophil extracellular trap, recurrence, subdural hematoma, vascular endothelial growth factor, VEGF

## Abstract

Non-acute subdural hematomas (NASHs) are a cause of significant morbidity and mortality, particularly with recurrences. Although recurrence is believed to involve a disordered neuroinflammatory cascade involving vascular endothelial growth factor (VEGF), this pathway has yet to be completely elucidated. Neutrophil extracellular traps (NETs) are key factors that promote inflammation/apoptosis and can be induced by VEGF. We investigated whether NETs are present in NASH membranes, quantified NET concentrations, and examined whether NET and VEGF levels are correlated in NASHs. Samples from patients undergoing NASH evacuation were collected during surgery and postoperatively at 24 and 48 h. Fluid samples and NASH membranes were analyzed for levels of VEGF, NETs, and platelet activation. NASH samples contained numerous neutrophils positive for NET formation. Myeloperoxidase-DNA complexes (a marker of NETs) remained elevated 48 h postoperatively (1.06 ± 0.22 day 0, 0.72 ± 0.23 day 1, and 0.83 ± 0.33 day 2). VEGF was also elevated in NASHs (7.08 ± 0.98 ng/mL day 0, 3.40 ± 0.68 ng/mL day 1, and 6.05 ± 1.8 ng/mL day 2). VEGF levels were significantly correlated with myeloperoxidase-DNA levels. These results show that NETs are increasing in NASH, a finding that was previously unknown. The strong correlation between NET and VEGF levels indicates that VEGF may be an important mediator of NET-related inflammation in NASH.

## 1. Introduction

Non-acute subdural hematomas (NASHs) are a heterogenous group of pathologies comprising subacute, chronic, and acute-on-chronic subdural hematomas (SDHs). In the general population, chronic SDH alone has an annual incidence of 10.35 per 100,000 [1], and this incidence increases significantly with age, with an annual incidence of 127.1 per 100,000 observed in the population 80 years of age and older [2]. NASHs are a significant cause of morbidity and mortality, owing to their high recurrence rates. Although the pathophysiological mechanism for recurrence is believed to involve multiple cell lines, chemo/cytokines, a disordered neuroinflammatory cascade involving vascular endothelial growth factor (VEGF), and pathological angiogenesis within the hematoma neomembrane, this pathway has not yet been completely elucidated [3,4,5,6]. NASHs have proven to be difficult to treat, with significant recurrence rates and implications for patients quality of life. Additionally, with the nation’s population of older adults increasing notably, the prevalence and impact of NASHs are likely to increase. Advances in the management of this disease are therefore more crucial than ever, and although surgical management has become more standardized and efficient and novel treatment modalities have been explored, a more thorough understanding of the pathophysiology of NASHs is necessary to continue to reduce the rate of recurrence.

Neutrophil extracellular traps, or NETs, are webs of decondensed chromatin, histones, and antimicrobial peptides/enzymes released by neutrophils [7]. Although they were originally described in the setting of acute infection for their antimicrobial properties, NETs have been shown to play an active role in sterile inflammatory processes such as sickle cell disease, small vessel vasculitis, deep vein thrombosis, and transfusion-related acute lung injury [8,9,10,11]. Recently, NETs have also been identified as key factors that interact with platelets and/or VEGF in acute ischemic stroke [12,13,14,15,16]. Given the importance of platelets and VEGF in both NASHs and ischemic strokes, it has been postulated that NETs may, by extension, play a crucial role in NASH pathophysiology. In an effort to further elucidate this role, the objectives of this study were threefold: (1) to demonstrate the presence of NETs in the membranes of NASHs; (2) to confirm the presence of NETs in evacuated NASH samples; and (3) to show a correlation between VEGF and NET levels in evacuated NASH samples.

## 2. Materials and Methods

### 2.1. Study Criteria

Institutional review board approval was obtained for this study. Informed consent was obtained either from the patient directly or from a legally authorized representative. Patients were enrolled in the study prospectively, with inclusion criteria including any patient ≥18 years with a NASH (chronic, subacute, or acute-on-chronic subdural hematoma) undergoing operative evacuation (via burr hole or miniature craniotomy) and subdural drain placement. Exclusion criteria consisted of patients with acute SDH, those under the age of 18 years, and patients being treated with solely nonoperative management. Healthy human donors older than 18 years from the Salt Lake City, Utah, area gave blood for plasma isolation after providing informed consent based on approval from the University of Utah IRB (#00051506).

### 2.2. Clinical Data

A total of 26 patients were enrolled in the study; clinical information was available for 11 of the 26 patients. Factors including age at time of surgery, the Glasgow Coma Scale (GCS) score, sex, preoperative medication usage (specifically regarding anticoagulation, antiplatelet agents, statins, and steroids), laterality of hematoma, recurrence requiring reoperation, in-hospital mortality, mortality at longest follow-up, and length of follow-up available were recorded.

### 2.3. Membrane Staining

NASH membranes were collected from patients intraoperatively whenever possible. Immediately after removal, membrane specimens were washed in 0.9% saline and fixed overnight in 4% paraformaldehyde. After a sucrose gradient, membrane specimens were snap-frozen in OCT compound and stored at 80 °C until cryosectioning into 10 µm slices. Before staining, slides were fixed in 4% paraformaldehyde and blocked in 3% donkey serum with 0.5% Tween 20. As primary antibodies, we used goat anti-myeloperoxidase (MPO) (2 µg/mL, AF3667, R&D Systems, Minneapolis, MN, USA), rabbit anti-human citrullinated histone H3 (2 µg/mL, ab5103, Abcam, Cambridge, UK), and mouse anti-GPIb (2 µg/mL, MA5-11642, Invitrogen, Waltham, MA, USA). As secondary antibodies, we used donkey anti-rabbit AF488, donkey anti-goat AF546, and donkey anti-mouse AF633 (Thermo Fisher, Waltham, MA, USA). DAPI (4′,6-diamidino-2-phenylindole) was used as a nuclear counterstain (Life Technologies, Carlsbad, CA, USA). Images were acquired using a high-resolution confocal reflection microscope (Olympus IX81, FV300, Melville, NY, USA).

### 2.4. NASH Sample Collection

NASH samples were collected from all enrolled patients at three timepoints: intraoperatively (day 0), 24 h postoperatively (day 1), and 48 h postoperatively (day 2) (via subdural drain). The blood was centrifuged at 2000× *g* for 10 min to remove red blood cells and other cellular components. Plasma from subdural hematomas was aliquoted and frozen for analysis at a later date.

### 2.5. MPO-DNA Complexes

An in-house enzyme-linked immunosorbent assay (ELISA) was used to quantify MPO-DNA complexes [16]. Briefly, after overnight coating with anti-MPO antibody (2 µg/mL; 0400-0002, Bio-Rad, Hercules, CA, USA) at 4 °C, a 96-well plate was blocked with 2.5% bovine serum albumin in phosphate-buffered solution for 2 h at room temperature. The plate was subsequently washed before incubating for 90 min at room temperature with 20% plasma isolated from the NASH in blocking buffer. The plate was washed five times and then incubated for 90 min at room temperature with an anti-DNA antibody (1:10; Cell Death Detection ELISA, 11544675001, Sigma, St. Louis, MO, USA). After five washes, the plate was developed with an ABTS substrate (Sigma).

### 2.6. Platelet Factor 4 and VEGF ELISAs

Commercially available ELISAs for platelet factor 4 (PF4, DPF400) and VEGF (DVE00) were from R&D Systems.

### 2.7. In Vitro NET Formation Using NASH Samples

Neutrophils were isolated from freshly collected whole blood of healthy adults using the EasySep Direct Human Neutrophil Isolation Kit (STEMCELL Technologies, Vancouver, Canada) with greater than 95% purity. Neutrophils were resuspended at a concentration of 1 × 106 cells/mL in plasma isolated from healthy donors or fluid from chronic SDH for one hour. After one hour, the neutrophils were washed and resuspended in M199 at 37 °C. After 2 h, cell-free media was isolated, and NETosis was measured using SYTOX Green DNA dye (ThermoFisher), a dye that measures cell-free DNA.

### 2.8. Statistical Analysis

Statistical analyses were performed with GraphPad Prism Version 9.1.2 (GraphPad Software, San Diego, CA, USA). All data are represented as dot plots, including a bar graph with error bars representing the mean ± standard error of the mean. A two-tailed *p* < 0.05 was considered statistically significant. Prior to analysis, a D’Agostino’s K^2^ test and a Pearson’s correlation coefficient test were used to check data distributions. A one-way ANOVA with Dunnett’s post-hoc test or Mann–Whitney tests, as appropriate, were used for statistical comparison when applicable. In the case of nonparametric data, Kruskal–Wallis tests with post-hoc Dunn’s correction were performed. Spearman’s rank correlation coefficient analysis was performed for all correlation studies.

## 3. Results

### 3.1. Clinical Data

Twenty-six patients were enrolled in the study and underwent NASH fluid collection. Clinical information could be obtained for 11 of the 26 patients (Table 1). The mean age was 71.8 ± 15.9 years, 72.7% were male (8 of 11), and the median GCS was 15. Three patients (27.3%) were on therapeutic anticoagulation (warfarin or apixaban), one (9.09%) was on prophylactic subcutaneous heparin, and two (18.2%) were taking aspirin. Preoperative statin use was noted in 45.5% (5 of 11) and 27.3% (3 of 11) were also treated with corticosteroids preoperatively. One patient had bilateral hematomas (9.09%), four had left-sided hematomas (36.4%), and the remaining six had right-sided hematomas (54.6%). One patient presented with a recurrence after a prior operation, but the remaining 10 had never had surgery, and none required repeat surgery. No patients sustained in-hospital mortality, but three patients died between 0.5 and 7 months after surgery, all from non-SDH-related illnesses. The average follow-up period was 9.5 ± 6.9 months.

### 3.2. NETs Are Present in Membranes from NASHs

When possible, the outer membrane of the NASH was procured intraoperatively. This membrane was examined for platelet and neutrophil deposition as well as NET formation (Figure 1). Confocal imaging revealed platelets (CD41+) and neutrophils (MPO+) were present. Interestingly, in regions where neutrophils were present, many neutrophils were positive for citrullinated histone H3 (H3Cit), a marker for NET formation. In addition, the triple colocalization of H3Cit, MPO, and DAPI, a DNA maker, strongly indicated that neutrophils were undergoing NET formation in the NASH membrane.

### 3.3. Markers of NET Formation Are Found in NASH Samples

We isolated fluid from NASHs from the 26 patients at the time of surgery and over the following 48 h (or until subdural drain removal) and examined markers of NET formation and platelet activation (PF4). Myeloperoxidase-DNA complexes (a marker of NETs) remained elevated 48 h postoperatively (1.06 ± 0.22 day 0, 0.72 ± 0.23 day 1, 0.83 ± 0.33 day 2). Seventeen of the 26 patients had NET levels based on MPO-DNA complexes above the NET levels of healthy donor plasmas (Figure 2a). NET levels appeared to decrease over time, with fewer MPO-DNA complexes seen on day 1. A small increase in NET levels was observed at day 2 compared with NET levels at day 1. Of the nine patients that had three serial samples, we observed a >20% decrease in NET formation in four patients and a >20% increase in NET levels in five patients. This effect was probably due to sampling bias, given that once patients had drains removed, they were discharged from the neurocritical care unit and further NASH sampling was not completed. 

Additionally, we examined markers of platelet activation in fluid from evacuated NASHs because platelets are known regulators of NET formation in the brain [16]. Although the platelet activation marker PF4 was present in some patients (Figure 2b), most patients had levels consistent with circulating plasma levels of PF4 in healthy donors.

### 3.4. VEGF Is Present in NASHs and Correlates with NET Level

We next examined whether VEGF might be present in NASHs. VEGF was elevated in NASHs compared to healthy controls. High levels of VEGF were found in the fluid from chronic SDH on day 0 and appeared to decrease on day 1 (7.08 ± 0.98 ng/mL on day 0 compared to 3.40 ± 0.68 ng/mL on day 1; Figure 2d). In samples from patients at day 2, we observed a slight increase over day 1 samples (6.05 ± 1.8 ng/mL at day 2 compared to 3.40 ± 0.68 ng/mL at day 1). 

We next investigated whether VEGF levels correlated with MPO-DNA complexes. VEGF levels are significantly (r = 0.51, *p* < 0.0001) correlated with NET formation, suggesting that VEGF may contribute to neutrophil activation and NET release in NASHs (Figure 3). MPO-DNA complexes did not correlate with platelet activation as measured by PF4 (Figure 3).

Finally, we incubated neutrophils with plasma from healthy donors or fluid from NASH samples to determine whether an agonist within the NASH samples was capable of inducing NETosis. NASH samples were dichotomized into those with low or high NETs to determine if the level of NETs present in the NASH sample was associated with the ability to induce NETosis. Consistent with a previous study, plasma from healthy donors did not induce NETosis (Figure 4) [17]. In contrast, fluid from NASH samples with high NET levels induced robust NET formation, while NASH samples with low NET levels did not induce NET formation above the healthy donor plasma control, suggesting that an agonist present in NASH induces NETosis (Figure 4). 

## 4. Discussion

In spite of a tremendous amount of research that has been dedicated to the study of NASHs, recurrence rates have not significantly decreased in recent years, with modern studies demonstrating rates of 15–17% [18,19]. Significant effort has been put into analyzing the factors that predict recurrence as well as optimizing surgery to reduce recurrence requiring reoperation; however, few changes have been made since closed drainage systems have demonstrated significant benefit [20,21,22]. This stagnation demonstrates that simply examining the utility of current treatments may be insufficient and that, rather, a better understanding of the pathophysiology of NASHs is necessary to improve outcomes in the treatment of this disease.

NASH chronicity and recurrence are believed to be a result of excess inflammation in the subdural microenvironment. After extravasation of blood into the subdural space, clot organization begins with the formation of a fibrous membrane around the hematoma, creating a distinct boundary between blood products and the area surrounding the brain. A significant inflammatory response is then elicited by the extravasated blood, which is mediated by proinflammatory chemokines and cytokines such as VEGF, hypoxia-inducible factor-1α (HIF-1α), interleukin-6 (IL-6), interleukin-8 (IL-8), basic fibroblast growth factor (bFGF), matrix metalloproteinases (MMPs), and tumor necrosis factor-α (TNF-α) [23,24,25,26]. Specifically, it has been demonstrated that neutrophils (which are in high concentration within NASH fluid) secrete MMPs and bFGF, which in turn cause cleavage of extracellular matrix proteins and the release of VEGF [24,27,28]. Although VEGF under normal conditions is responsible for physiologic angiogenesis, excessive VEGF levels can inhibit vascular maturation, resulting in friable, leaky blood vessels that are prone to hemorrhage [4,24,29]. NETs, which are chromatin webs extruded from neutrophils, have been found at elevated levels in acute ischemic stroke [16,30,31,32]. Additionally, NET formation has been significantly associated with VEGF expression across a variety of pathologies and has been shown to cause pathological angiogenesis [33,34,35]. Given that NASH formation and recurrence are believed to be related to VEGF-induced pathological angiogenesis [26,36,37], we hypothesized that NETs play an important role in the pathophysiology of NASHs. We found that NETs were present in NASH membranes, as indicated by H3Cit+ neutrophils on confocal imaging. Strong NET production within NASH membranes may indicate a vicious cycle in which the source for elevated NETs in NASHs are the leaky and friable vessels within NASH membranes, which are proposed to be the source of chronic microbleeding in recurrent NASHs, which then leads to further dysregulated angiogenesis within the NASH [38,39,40,41]. 

NASH fluid was also found to have greater levels of NET formation compared with control plasma, and NASH fluid was remarkably capable of inducing NETosis in control plasma. The levels of platelet activation, however, were similar to those of control plasma. This would suggest that while NETs are in fact present and elevated in NASH fluid, platelet activation does not appear to be related to their increased levels. Although platelet dysregulation is a known mediator of NETosis, this finding suggests that a platelet-independent pathway of NET formation exists in NASH. 

Lastly, VEGF and NET levels were found to be significantly correlated in NASH samples (*p* < 0.0001). Given that VEGF is known to induce NETosis in other pathologies, it likely plays a key role in inducing NET formation in NASHs as well [33]. Thus, VEGF-mediated NETosis may represent either an independent pathway or a potential regulator in the already proposed multifactorial inflammatory cascade of cytokines/chemokines that lead to pathological angiogenesis in NASH formation and recurrence. 

The main limitation of this study is the sample size, which may explain certain findings such as the uptrend in NET and VEGF on day 3 (although this may also be attributable to the delayed reaccumulation of blood after drainage and therefore the resumption of the chronic microbleeding cycle). Additional limitations include the fact that in order for NASH fluid to be analyzed, the patients had to undergo surgery, which consequently excluded NASHs that were not evacuated. In turn, one might posit that the pathophysiology of smaller NASHs that do not undergo operative intervention might involve a less dysregulated inflammatory state. In addition, because of the limited sample size, the effect of antiplatelet medications or other medications such as statins, which could affect NET formation and platelet activation, cannot be precisely assessed.

## 5. Conclusions

Our pilot study demonstrates that NETs are present in both NASH samples and NASH membranes obtained at the time of surgery. There is a significant correlation between NET levels and VEGF levels within NASH samples, indicative of an active relationship between these two factors within the SDH fluid itself. This finding highlights a novel pathway in the inflammatory cascade believed to be the cause of NASH recurrence and offers another key point at which interventions, such as NET inhibitors, may be of use in reducing inflammatory changes and subsequent pathological angiogenesis. Further studies are warranted to examine the exact role of NETs in NASHs and to examine the possibility of interventions that may help reduce the frequency of NASH recurrence.

## Figures and Tables

**Figure 1 diagnostics-12-02934-f001:**
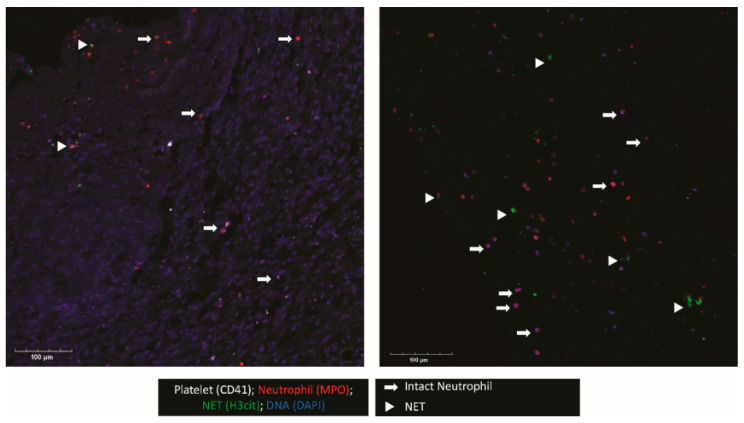
NETs and platelets are present in NASH outer membranes. The outer membranes of NASHs were collected, fixed, and embedded in OCT. Membranes were cryosectioned and stained for platelets (CD41, white), neutrophils (MPO, red), and H3Cit, a marker of NETosis (green). Nucleated cells were stained with DAPI (blue). The image shown is representative of three independent samples.

**Figure 2 diagnostics-12-02934-f002:**
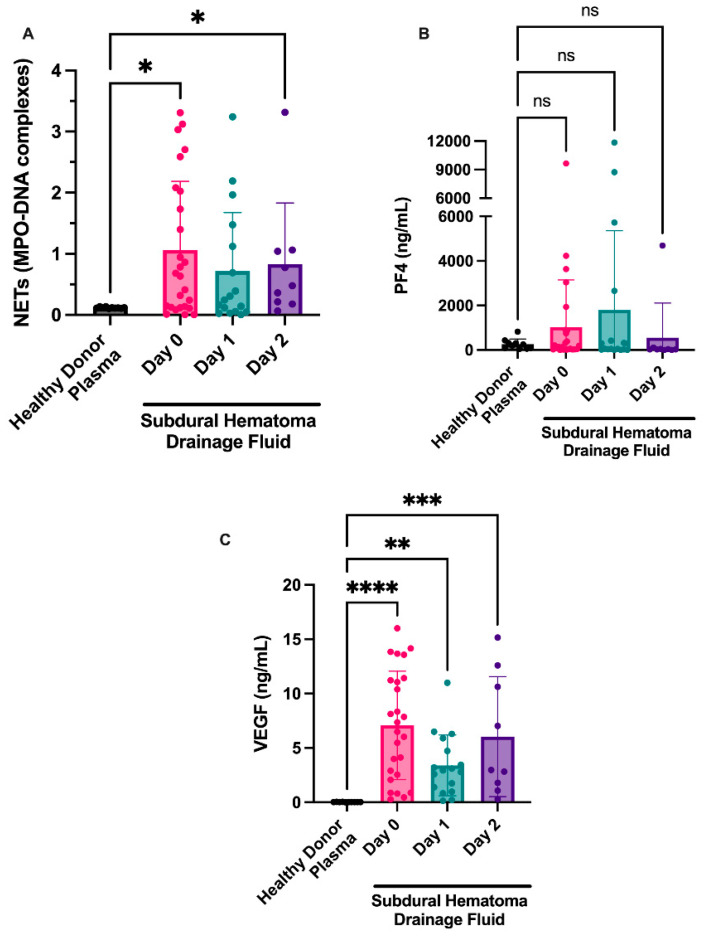
NETosis is observed in fluid from NASHs. NASH samples were collected from patients intraoperatively (day 1) and from subdural drains on the indicated days. Cellular components were removed as described in the Methods section. NETosis was measured by the presence of MPO-DNA complexes (**A**). As a comparison, NET, PF4, and VEGF levels were examined in 10 healthy donors. Platelet activation was measured by a commercially available ELISA for PF4 (**B**). VEGF was measured in the fluid from NASH using an ELISA (**C**). As a comparison, PF4 and VEGF levels were examined in 10 healthy donors. * *p* ≤ 0.05, ** *p* ≤ 0.01, *** *p* ≤ 0.001, and **** *p* ≤ 0.0001.

**Figure 3 diagnostics-12-02934-f003:**
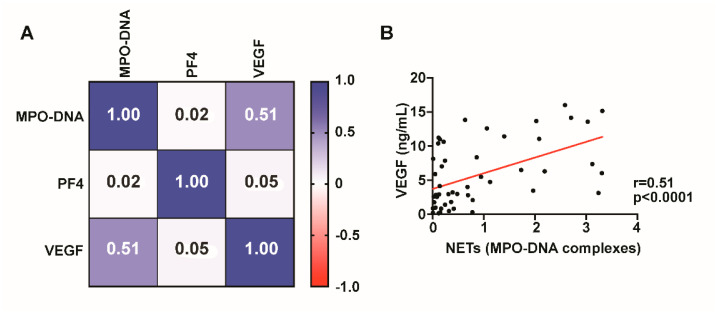
NET formation correlates with VEGF levels. Spearman’s rank correlation coefficient matrix of NET levels compared with PF4 and VEGF using day 1 CDSH fluid (**A**). MPO-DNA complexes significantly (*p* < 0.0001) correlate with VEGF levels in fluid from NASH ((**B**), *p* < 0.0001).

**Figure 4 diagnostics-12-02934-f004:**
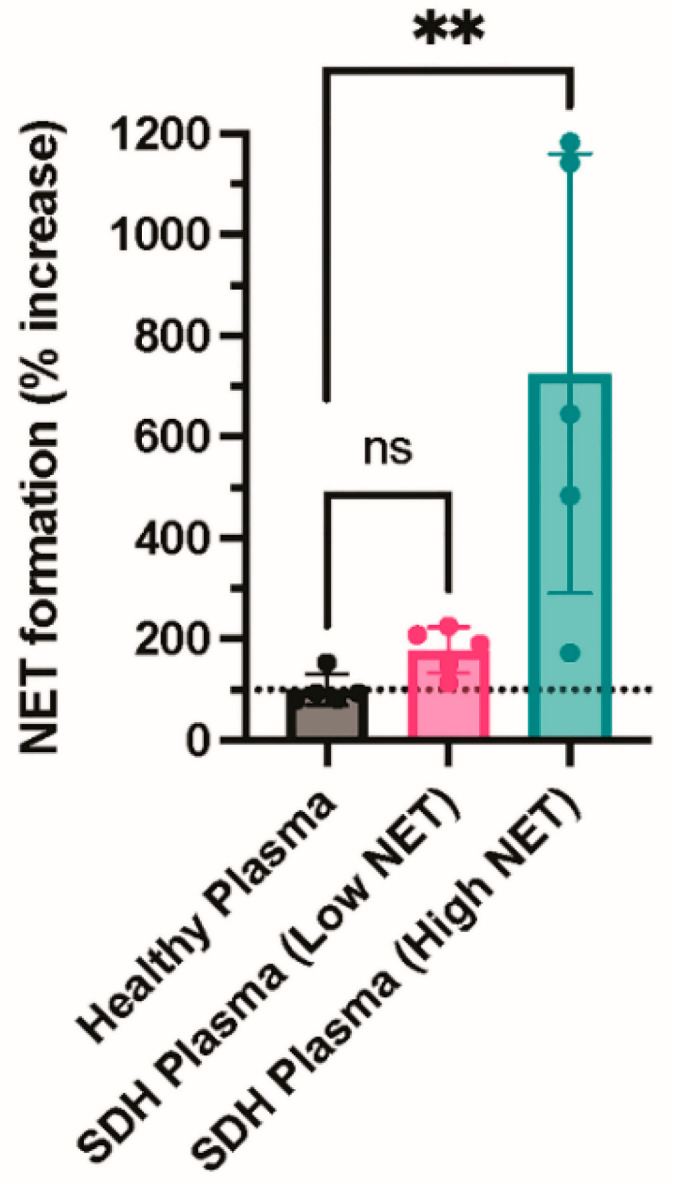
NASH fluid increases NETosis in healthy neutrophils compared with healthy plasma. Neutrophils were isolated from healthy donors using CD15+ selection. Neutrophils were incubated with buffer (resting), healthy plasma, or fluid from NASH samples for one hour. After one hour, neutrophils were resuspended in M199 media and allowed to undergo NETosis. Cell-free media was collected, and NET formation was measured by cell-free DNA release (N = 5 per group). ** *p* ≤ 0.01. ns= not significant.

**Table 1 diagnostics-12-02934-t001:** Demographic and clinical information for 11 of the 26 patients enrolled.

*Patient No.*	*Age (Years)/Sex*	*GCS Score*	*Anticoagulation*	*Aspirin*	*Preoperative*		*Laterality*	*Recurrence* *Requiring* *Reoperation*	*Mortality ^1^*	*Follow-Up (Months)*
					Statin	Steroid				
*1*	64/F	14	Prophylactic heparin	No	No	No	Right	Hx prior operation	-	2
*2*	85/M	9	None	Yes	Yes	No	Right	No	Leukemia	2
*3*	66/M	15	None	Yes	Yes	Yes	Bilateral	No	-	5
*4*	77/M	13	Warfarin	No	No	No	Left	No	-	16
*5*	43/M	15	None	No	Yes	Yes	Right	No	-	18
*6*	89/M	15	Apixaban	No	Yes	No	Right	No	-	12
*7*	62/M	14	None	No	No	No	Right	No	Pneumatosis intestinalis	0.5
*8*	55/F	15	None	No	No	No	Right	No	-	19
*9*	91/M	15	None	No	No	No	Left	No	-	18
*10*	95/F	11	None	No	No	No	Left	No	Urinary infection	7
*11*	63/M	15	Warfarin	No	Yes	Yes	Left	No	-	5

GCS = Glasgow Coma Scale; Hx, history; ^1^ None were in-hospital.

## Data Availability

All data are available from the corresponding authors.

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
