# Peer review of "Investigation of Neutrophil Extracellular Traps as Potential Mediators in the Pathogenesis of Non-Acute Subdural Hematomas: A Pilot Study"

_diagnostics, 2022, doi:10.3390/diagnostics12122934_

Round 1

Reviewer 1 Report

In this study, the authors aim to investigate the relationship between the neutrophil extracellular traps (NETs) and vascular endothelial growth factor (VEGF) in nonacute subdural hematomas (NASHs). While the level of NETs and VEGF may be the cause of NASH, the reviewer has several major concerns that render this manuscript not suitable for publication in MDPI diagnostics:

NASH is a frequent disease in the elder people. However, the patients number in this study was not enough, which render the results of the present study not convincing.

The inclusion and exclusion criteria should be explained in detail.

In addition, it is also necessary to specify what type of surgery was performed and whether the patients were treated in the same method.

I’m interested in the recurrence rate in these patients. In the conclusion, it was indicated that NET may be a novel pathway in the inflammatory cascade, which was believed to be the cause of NASH recurrence. Please provide more details.

References need to be updated. It is recommended that authors should demonstrate more new references according to the decreasing recurrence rates.

Whether the preoperative medication (e.g. steroid and anti-coagulation) has an effect on the experimental results, such as stain.

Author Response

In this study, the authors aim to investigate the relationship between the neutrophil extracellular traps (NETs) and vascular endothelial growth factor (VEGF) in nonacute subdural hematomas (NASHs). While the level of NETs and VEGF may be the cause of NASH, the reviewer has several major concerns that render this manuscript not suitable for publication in MDPI diagnostics:

NASH is a frequent disease in the elder people. However, the patients number in this study was not enough, which render the results of the present study not convincing.

  • Although we appreciate that a greater sample size would be ideal, the nature of this study is that of a pilot study to primarily demonstrate the presence of a novel factor (NETs) in SDH. This initial sample is therefore a starting point from which we may base further investigation. That it is a pilot study has now been added to the title in response to reviewer 3.

 The inclusion and exclusion criteria should be explained in detail.

  • Inclusion and exclusion criteria are discussed on page 2, paragraph 2, with additional details now added to this section.

 In addition, it is also necessary to specify what type of surgery was performed and whether the patients were treated in the same method.

  • Specification of surgery performed (burr hole or miniature craniotomy) has been added to page 2, paragraph 2.

 I’m interested in the recurrence rate in these patients. In the conclusion, it was indicated that NET may be a novel pathway in the inflammatory cascade, which was believed to be the cause of NASH recurrence. Please provide more details.

  • Further explanation for the inflammatory etiology of NASH has been provided on page 10, paragraph 2.

“NASH chronicity and recurrence is believed to be a result of excess inflammation in the subdural microenvironment. After extravasation of blood into the subdural space, clot organization begins with the formation of a fibrous membrane around the hematoma, creating a distinct boundary between blood products and surrounding brain. A significant inflammatory response is then elicited by the extravasated blood, mediated by proinflammatory chemokines and cytokines including VEGF, hypoxia-inducible factor-1α (HIF-1α), interleukin (IL)-6, IL-8, basic fibroblast growth factor (bFGF), ma-trix metalloproteinases (MMPs), and tumor necrosis factor-α (TNF-α) [23-26]. Specifically, it has been demonstrated that neutrophils (which are in high concentration within NASH fluid) secrete MMPs and bFGF, which in turn cause cleavage of extra-cellular matrix proteins and the release of VEGF [24,27,28]. Although VEGF under normal conditions is responsible for physiologic angiogenesis, excessive VEGF levels can inhibit vascular maturation, resulting in friable, leaky blood vessels that are prone to hemorrhage [4,24,29].”

 References need to be updated. It is recommended that authors should demonstrate more new references according to the decreasing recurrence rates.    

  • Updated recurrence rates and references have been added to page 7, paragraph 1. The revised text now states “In spite of a tremendous amount of research that has been dedicated to the study of NASHs, recurrence rates have not significantly decreased in recent years, with modern studies demonstrating rates of 15-17% [18,19].”

 Whether the preoperative medication (e.g. steroid and anti-coagulation) has an effect on the experimental results, such as statin.

  • The sample size of our current study is too limited to draw strong conclusions about the impact of medications on the experimental results. We have included a sentence in the discussion (p. 8) stating this is a limitation.

“In addition, because of the limited sample size, the effect of antiplatelet medications or other medications such as statins, which could affect NET formation and platelet activation, cannot be precisely assessed.”

Reviewer 2 Report

Comment to authors (Diagnostics; 1954497)

Authors analyzed correlation between neutrophil extracellular traps (NET) and vascular endothelial growth factor (VEGF) in nonacute subdural hematomas (NASHs).

Those parameters might represent pathogenesis and pathophysiology of NASHs and be expected as prediction and prognostic biomarkers for recurrence of NASHs.

This is an interesting paper. I think the paper is worth to be published.

I have several questions to be addressed.

#1. Authors collected blood samples from normal volunteers, which should be important to confirm whether these inflammatory reactions would be specific to NASHs. Therefore, levels of platelet factor 4 (PF4) and VEGF should be presented to compare with those in patients with NSDH in figure 2C and 2D.

#2. I am very curious about levels of PF4, VEGF, and NET in recurrent NASHs compared with newly diagnosed NASHs. Did authors have an opportunity to analyze levels of PF4, VEGF, and NET in same patients with both newly diagnosed and recurrent NASHs? (Although, it said “none required repeat surgery” in the paper.)

#3. Based on figure 2C, level of PF4 seemed to be influenced by surgical insults and wound healing process including hemostasis with its peak level on day 2. What about antiplatelet agents? Were levels of PF4 in patients with and without antiplatelet administration?

#4. I think that steroid and statin administration might have any impacts on levels of PF4, VEGF, and NET. Some proposed that statin might have an effect on prophylaxis of chronic subdural hematoma. How about that?

Author Response

Authors analyzed correlation between neutrophil extracellular traps (NET) and vascular endothelial growth factor (VEGF) in nonacute subdural hematomas (NASHs).

Those parameters might represent pathogenesis and pathophysiology of NASHs and be expected as prediction and prognostic biomarkers for recurrence of NASHs.

This is an interesting paper. I think the paper is worth to be published.

I have several questions to be addressed.

#1. Authors collected blood samples from normal volunteers, which should be important to confirm whether these inflammatory reactions would be specific to NASHs. Therefore, levels of platelet factor 4 (PF4) and VEGF should be presented to compare with those in patients with NSDH in figure 2C and 2D.

  • In updated figures, we now compare PF4 and VEGF levels to those with NSDH. Please see updated Figure 2.

#2. I am very curious about levels of PF4, VEGF, and NET in recurrent NASHs compared with newly diagnosed NASHs. Did authors have an opportunity to analyze levels of PF4, VEGF, and NET in same patients with both newly diagnosed and recurrent NASHs? (Although, it said “none required repeat surgery” in the paper.)

  • We were unable to answer this question because of the small sample size of our patient cohort. We are actively investigating this question currently in the laboratory.

#3. Based on figure 2C, level of PF4 seemed to be influenced by surgical insults and wound healing process including hemostasis with its peak level on day 2. What about antiplatelet agents? Were levels of PF4 in patients with and without antiplatelet administration?

  • Unfortunately, only 2 of the patients that we had clinical information on were on aspirin, making it difficult to draw a conclusion about the role of antiplatelet agents altering PF4 levels. We are actively investigating this in the laboratory currently.

#4. I think that steroid and statin administration might have any impacts on levels of PF4, VEGF, and NET. Some proposed that statin might have an effect on prophylaxis of chronic subdural hematoma. How about that?

  • A previous study suggested statins may enhance NET release (Chow OA et al Cell Host Microbe 2010), while another study suggested LDL apheresis through dextran sulfate reduced circulating PF4 levels (Tanhehco YC et al Transfusion 2011). Although previous literature supports the idea that statins and lowering LDL levels can alter PF4 levels and NET formation, the number of patients in our current cohort is too small to make any definitely conclusions about the effect of statins in our patients.

Reviewer 3 Report

The aim of the publication entitled: „Investigation of Neutrophil Extracellular Traps as Potential Mediators in the Pathogenesis of Nonacute Subdural Hematomas” was to investigated, whether NETs are present in NASH membranes, quantifie NET concentrations, and examine whether NET and VEGF levels are correlated in NASHs.

The manuscript is interesting and may be helpful to explaining the pathophysiology of NASH, and in the long term, also in finding appropriate pharmacotherapy.

However, I have comments that will be necessary to improve the manuscript.

1.           The title should contain the information "pilot study"

2.           The aim of the study needs to be supplemented, why were the platelets stained in the study and the platelet activation markers were analyzed?

3.           The performed statistical analysis raises doubts and requires correction. Results are shown as mean ± standard deviation, what tests were used? This should be added.

4.           Due to the small size of the study group, I suggest that Figures 2 B, C, D are presented as dot plot.

5.           The units in the 2D figure are wrong, please correct it.

6.           The results section needs to be sorted out. Information explaining the purpose of the research, e.g.

"Because platelets are known to play a key role in NET formation, we examined markers of platelet activation in fluid from evacuated NASHs".

" ... Dysregulated platelet activation was not implicated in the pathophysiology of NETosis in NASH [16]."

should be moved to the introduction/discussion.

7.           The sentence  from discussion section: "In spite of a tremendous amount of research that has been dedicated to the study of NASHs, recurrence rates have not significantly decreased in recent years" requires the addition of an appropriate references.

8.           This fragment of: Conclusions section: "NASHs have proven to be difficult to treat, with significant recurrence rates and implications to patients' quality of life. Additionally, with the nation's population of older adults increasing notably, the prevalence and impact of NASHs is likely to increase. Advances in management of this disease are therefore more crucial than ever, and while surgical management has become more standardized and efficient, and novel treatment modalities have been explored, a more thorough understanding of the pathophysiology of NASHs is necessary to continue to reduce the rate of recurrence " does not constitute conclusions of the obtained results, it may supplement the introduction/discussion.

Author Response

The aim of the publication entitled: „Investigation of Neutrophil Extracellular Traps as Potential Mediators in the Pathogenesis of Nonacute Subdural Hematomas” was to investigated, whether NETs are present in NASH membranes, quantifie NET concentrations, and examine whether NET and VEGF levels are correlated in NASHs.

The manuscript is interesting and may be helpful to explaining the pathophysiology of NASH, and in the long term, also in finding appropriate pharmacotherapy.

However, I have comments that will be necessary to improve the manuscript.

  1. The title should contain the information "pilot study"
  • This has been added to the title.
  1. The aim of the study needs to be supplemented, why were the platelets stained in the study and the platelet activation markers were analyzed?
  • The aims of the study are included on page 2, paragraph 1. Previous study has suggested that platelets are key regulators of NET formation. To examine whether platelets could contribute to NET formation in NASH, we examined whether platelets were present and activated during NASH.

“In an effort to further elucidate this role, the objectives of this study were threefold: 1) to demonstrate the presence of NETs in the membranes of NASHs; 2) to confirm the presence of NETs in evacuated NASH samples; and 3) to show a correlation between VEGF and NET levels in evacuated NASH samples.”

  1. The performed statistical analysis raises doubts and requires correction. Results are shown as mean ± standard deviation, what tests were used? This should be added.
  • Details of statistical analysis are included on page 3, paragraph 6.

“All data are represented as dot plots including a bar graph with error bars representing mean ± standard error of the mean. A two-tailed p<0.05 was considered statistically significant. Prior to analysis, a D’Agostino and Pearson normality test was used to check data distributions. One-way ANOVA with Dunnett’s post-hoc test or Mann-Whitney tests, as appropriate, were used for statistical comparison when applicable. In the case of nonparametric data, Kruskal–Wallis tests with post-hoc Dunn correction were performed. Spearman correlation analysis was performed for all correlation studies.”

  1. Due to the small size of the study group, I suggest that Figures 2 B, C, D are presented as dot plot.
  • Figure 2A, B, and C are now presented as dot plots.
  1. The units in the 2D figure are wrong, please correct it.
  • We apologize for the wrong units in Figure 2D. This has been corrected. Please note in the revised manuscript this is Figure 2C
  1. The results section needs to be sorted out. Information explaining the purpose of the research, e.g.

"Because platelets are known to play a key role in NET formation, we examined markers of platelet activation in fluid from evacuated NASHs".

" ... Dysregulated platelet activation was not implicated in the pathophysiology of NETosis in NASH [16]."

should be moved to the introduction/discussion.

  • These and similar statements have been removed from the Results.
  1. The sentence  from discussion section: "In spite of a tremendous amount of research that has been dedicated to the study of NASHs, recurrence rates have not significantly decreased in recent years" requires the addition of an appropriate references.
  • References with recent recurrence levels have been added to paragraph 1 of the Discussion.
  1. This fragment of: Conclusions section: "NASHs have proven to be difficult to treat, with significant recurrence rates and implications to patients' quality of life. Additionally, with the nation's population of older adults increasing notably, the prevalence and impact of NASHs is likely to increase. Advances in management of this disease are therefore more crucial than ever, and while surgical management has become more standardized and efficient, and novel treatment modalities have been explored, a more thorough understanding of the pathophysiology of NASHs is necessary to continue to reduce the rate of recurrence " does not constitute conclusions of the obtained results, it may supplement the introduction/discussion.
  • This section has been moved to the Introduction.

Round 2

Reviewer 1 Report

Thanks for the responses. However, I’m still confused with the number of the patients. How did you enroll the 26 patients and choose the 11 of them?  Please provide the starting time of case enrollment.

The sentence in result section: " Seventeen of the 26 patients had NET levels based on MPO-DNA complexes above the NET levels of healthy donor plasmas (Figure 2A). " . It requires explanation about the seventeen patients instead of eleven patients.

Author Response

Thanks for the responses. However, I’m still confused with the number of the patients. How did you enroll the 26 patients and choose the 11 of them? Please provide the starting time of case enrollment.

RESPONSE: There were a total of 26 patients enrolled in the study and samples were analyzed from all 26 patients. However, we were only able to obtain all the necessary clinical information on 11 of the 26 patients. This information is presented in Table 1. We have modified the title of Table 1 and the Methods section entitled Clinical Data to reflect that only 11 of the 26 patients had all the clinical data available.

The sentence in result section: " Seventeen of the 26 patients had NET levels based on MPO-DNA complexes above the NET levels of healthy donor plasmas (Figure 2A). " . It requires explanation about the seventeen patients instead of eleven patients.

RESPONSE: As described above, 26 patients were enrolled in the study and had NASH fluid collected and analyzed as shown in Figure 2. However, only 11 of the 26 patients had clinical information available to report in the manuscript. We have clarified this in the text in the Methods section, Table 1, and the Results section.